# Short-Term vs. Long-Term: A Critical Review of Indoor Radon Measurement Techniques

**DOI:** 10.3390/s24144575

**Published:** 2024-07-15

**Authors:** Khathutshelo Vincent Mphaga, Thokozani Patrick Mbonane, Wells Utembe, Phoka Caiphus Rathebe

**Affiliations:** 1Department of Environmental Health, Faculty of Health Sciences, Doornfontein Campus, University of Johannesburg, P.O. Box 524, Johannesburg 2006, South Africa; tmbonane@uj.ac.za (T.P.M.); wutembe@cartafrica.org (W.U.); prathebe@uj.ac.za (P.C.R.); 2National Health Laboratory Service, Toxicology and Biochemistry Department, National Institute for Occupational Health, Johannesburg 2000, South Africa

**Keywords:** indoor radon, short-term radon testing (STT), long-term radon testing, Europe, United States (US)

## Abstract

Radon is a known carcinogen, and the accurate assessment of indoor levels is essential for effective mitigation strategies. While long-term testing provides the most reliable data, short-term testing (STT) offers a quicker and more cost-effective alternative. This review evaluated the accuracy of STT in predicting annual radon averages and compared testing strategies in Europe (where long-term measurements are common) and the United States (where STT is prevalent). Twenty (20) studies were systematically identified through searches in scientific databases and the grey literature, focusing on STT accuracy and radon management. This review revealed several factors that influence the accuracy of STT. Most studies recommended a minimum four-day test for initial screening, but accuracy varied with radon levels. For low levels (<75 Bq/m^3^), a one-week STT achieved high confidence (>95%) in predicting annual averages. However, accuracy decreased for moderate levels (approximately 50% success rate), necessitating confirmation with longer testing periods (3 months). High radon levels made STT unsuitable due to significant fluctuations. Seasonality also played a role, with winter months providing a more representative picture of annual radon averages. STT was found to be a useful method for screening low-risk areas with low radon concentrations. However, its limitations were evident in moderate- and high-level scenarios. While a minimum of four days was recommended, longer testing periods (3 months or more) were crucial for achieving reliable results, particularly in areas with potential for elevated radon exposure. This review suggests the need for further research to explore the possibility of harmonizing radon testing protocols between Europe and the United States.

## 1. Introduction

There has been growing concern about Indoor Air Quality (IAQ) due to the recognition that people spend most of their time indoors [1,2,3]. Highlighting these concerns, international organizations like United Nations Scientific Committee on the Effects of Atomic (UNSCEAR) emphasize the health risks posed by indoor air pollutants, particularly radon [4]. This invisible, tasteless and odorless gas seeps into buildings from the ground, posing a significant health risk. Radon is the second leading cause of lung cancer for smokers and the leading cause for non-smokers [5]. According to WHO data, 3 to 14% of lung cancer cases worldwide are attributed to indoor radon [5,6], while UNSCEAR estimated that radon causes 1 in 10 lung cancer cases [7,8]. International organizations (IAEA, EPA, WHO, ICRP) recommend reference levels for indoor radon exposure (100–300 Bq/m^3^) to protect public health [5,6]. Recent studies suggest that there is no safe threshold for indoor radon exposure, as its carcinogenic effects have been reported below the established reference levels [5,9]. The adopted reference level (RL) or action level (AL) for indoor radon vary by region due to geological differences, necessitating flexible national frameworks [1,10,11].

The concentration of indoor radon is affected by geological formations, environmental factors, building construction (including materials and ventilation systems), and occupant behavior. Predicting indoor radon levels based solely on these factors is unreliable, underscoring the importance of direct indoor measurements for accurate risk assessment and targeted mitigation efforts [12]. Since radon toxicity does not present with warning signs or symptoms, measuring and keeping indoor radon levels low is essential [13]. Continuous indoor radon monitoring is crucial due to conclusive scientific evidence linking indoor radon exposure to lung cancer [14]. International radiation agencies (such as IAEA and ICPR) advocate for public awareness campaigns on radon, routine radon testing, and the implementation of effective mitigation strategies to address this public health concern [15,16,17]. 

Radon testing methods vary in terms of duration. Long-term measurements typically use passive detectors (i.e., nuclear track detectors) for three months, while short-term testing (STT) often uses active monitors to achieve quicker results—within one to seven days [5,9,11,14,18]. The trade-off between rapid results and long-term accuracy in radon measurement is highlighted by the distinction between passive and active methods. Long-term measurements are essential for making accurate annual average concentration estimates due to the correlation between lung cancer risk and cumulative radon exposure [19]. STT remains valuable for rapid screening (e.g., real estate transactions) or validating radon indoor radon levels in new buildings and when long-term measurement is impractical [19]. This review assessed the precision of STT in forecasting annual indoor radon averages. We compared European long-term measurement strategies with the US’s prevalence of STTs, acknowledging the lack of global standardization [18]. While long-term radon measurement techniques are the gold standard, their resource-intensiveness necessitates the exploration of alternatives. This investigation focused on the trade-off between STT simplicity/cost and associated uncertainty. Our core question is whether STTs provide sufficiently accurate data to inform public health strategies to mitigate indoor radon exposure. We critically evaluated the viability of STTs as a predictor of annual averages within the context of public health applications. Furthermore, we compared radon management strategies used in Europe to those that are common in the US.

## 2. Materials and Methods

### Search Strategy and Selection Criteria

We conducted a systematic literature search across various electronic databases (PubMed, Google Scholar) and grey literature sources (organizational websites such as US EPA, WHO, European Commission, IAEA). Furthermore, we scrutinized the reference lists of relevant articles. Our search strategy employed a combination of keywords and MeSH terms to capture all pertinent studies: indoor radon AND (short-term measurement OR long-term measurement Indoor radon AND (United States OR Europe OR US OR EU). We included studies written in English regardless of publication date. Following the PRISMA framework, we identified relevant studies through a multi-stage screening process. The initial search in PubMed and Google Scholar yielded 473 articles (433 + 40). After adding 10 records from agency websites and reference lists, a total of 483 articles were screened. Ultimately, 20 studies met the inclusion criteria and were selected for this critical review. A comprehensive search strategy, following the PRISMA framework, is outlined in Figure 1 below.

## 3. Results

### 3.1. Best Practices in Radon Measurement 

The studies summarized in Table 1 offered valuable insights into the best practices in radon measurement. These studies explored a range of key areas, including policy and regulations, community engagement initiatives, and standardized practices. The table also highlights successful the strategies employed to raise public awareness and encourage radon testing in residential settings. Examples include the collaborative efforts undertaken in Iowa [20], and the community involvement initiatives implemented in Canada [21].

An analysis of radon testing practices across different regions highlighted key factors for successful implementation, as shown in Table 1. Community involvement and financial incentives have been found to significantly increase testing rates, as demonstrated by a Canadian program’s 97% response rate. Additionally, collaborative partnerships between various stakeholders, such as public health and NGOs, have proven to be highly effective. Extensive public awareness campaigns that use multiple media channels have been crucial in driving behavioral change. Finally, standardized testing methodologies, such as the tiered approach with short-term and long-term measurements, have ensured accurate assessments. Therefore, a comprehensive strategy that integrates these elements offers the most promising approach to improving radon testing rates and ultimately safeguarding public health. 

### 3.2. Short-Term Radon Measurement Accuracy

In the field of radon exposure assessment, it is essential to mitigate the health risks associated with this radioactive gas. The decision to mitigate risks is often based on indoor radon measurement readings. While continuous long-term monitoring is effective, it can be costly and time-consuming. As a result, short-term radon detectors have gained attention as potentially attractive alternatives [22]. Table 2 delves into the effectiveness of STT in predicting long-term exposure levels.

**Table 1 sensors-24-04575-t001:** Radon management best practices.

Author, Year	Key Findings
Casey et al., 2018	This article explored a three-year program in Canada, focusing on testing residential areas for radon over a period of 91 days. Achieving a 97% response rate, the project’s success was attributed to community involvement, financial support, and increased awareness. Consequently, fewer homes exhibited radon levels surpassing the reference limit of 200 Bq/m^3^. Additionally, this paper highlighted the activities and factors that contributed to the project’s success [20].
Gordon et al., 2018	This study examined radon regulations in US schools, focusing on identifying key policy features and inconsistencies. Despite mandatory radon testing in most US schools, state regulations varied. While test results were usually disclosed, trained professionals typically conducted testing, which some schools struggled to afford. To ease the financial burden, some states (e.g., Indiana, Maine) offer testing cost support. Experts recommend a two-step approach for increased testing frequency, with initial school-led tests followed by professional confirmation [23].
Bain et al., 2016	This report provides details on strategies that led to increased radon testing in Iowa between 2010 and 2015. Citing a 42% lung cancer link to radon, the state implemented a collaborative approach with diverse partners (public health, NGOs, healthcare, etc.) to raise awareness and testing. Registered professionals ensured quality testing/mitigation, while media campaigns (including social media and Youtube) promoted radon education. These efforts resulted in a 20% increase in radon testing in Iowa [21].
Lee et al., 2016	A comparative analysis of radon measurement techniques in Canada and the US revealed a shared reliance on a tiered approach, albeit with variations in test duration. Typically, an initial radon test was conducted for a minimum of three months, with a year-long measurement conducted if the reference limit was exceeded. Meanwhile, in the UK, passive radon detectors are commonly used for long-term measurements to validate radon levels in homes that exceed the recommended values set by the government [24].

**Table 2 sensors-24-04575-t002:** Short-term test (STT) accuracy in predicting long-term (annual) radon exposure.

Author, Year	Purpose	Methodology	Findings
Warkentin et al. 2020	Assessed the accuracy of consumer-grade radon monitors.	The Radiation Safety Institute of Canada (RSIC) assessed consumer radon detectors (e.g., Radon Eye Plus) in a controlled chamber, comparing their readings to a reference monitor (AlphaGuard).	Consumer-grade radon monitors exhibited lower accuracy at levels near the recommended limit (200 Bq/m^3^) and performed better in winter compared to summer [25].
2. Warkentin et al. 2015	Assessed the accuracy of short-term (5-day) and medium-term (30-day) radon testing compared to a long-term (91-day) test.	During the heating season in Canada, electret ion radon detectors were employed to assess radon levels in 50 homes across three testing periods (5 days, 30 days, and 91 days).	One-week tests were accurate for low-risk areas/new buildings, but high-risk zones required 3 months [26].
3. Groves-Kirkby et al., 2006	To identify the most effective integration-based detector technology for short-term domestic radon assessment.	Thirty-four high-radon homes were monitored using track–etch detectors for 1–3 months alongside one-week co-located track–etch, charcoal, and electret detectors deployed monthly. Three homes additionally used continuous radon monitors for extended periods.	Analysis showed one-week track–etch, charcoal, and electret results below 75 Bq/m^3^ guaranteed (with 95% confidence) that an annual average was under the UK’s 200 Bq/m^3^ action level [27].
4. Miles et al., 2004	Assessed the reliability and accuracy of CR39 radon detectors	The UK implemented a validation scheme for radon track–etch detector labs, requiring adherence to proper procedures and accurate reporting. Labs undergo performance tests every six months to ensure they meet specified uncertainty levels.	Short-term tests (<75 Bq/m^3^) reliably predicted annual averages below the action level (200 Bq/m^3^), while high results (>75 Bq/m^3^) required longer testing periods. Values exceeding 500 Bq/m^3^ suggested exceeding the action level [28].
5. Janik et al., 2012	To develop a methodology for estimating the uncertainty associated with using short-term radon measurements to predict long-term radon averages.	Radon concentrations were measured in dwellings, cellars, and outdoor environments using AlphaGUARD devices. Measurement durations ranged from 16 days to a year, with data collected every 10 or 60 min. Notably, a minimum total measurement time of 16 days was identified as essential for robust data analysis.	Combining field data and models established a minimum 4-day measurement time for reliable radon screening, minimizing short-term measurement errors [29].
6. Arafa et al., 1994	Investigated the suitability of the activated charcoal technique with two scintillation detectors to measure short-term radon concentrations in Qatar.	Activated charcoal canisters measured gamma rays from radon daughters. A calibration factor was obtained using a reference canister with known radon exposure. A computer program optimized exposure time to enhance accuracy and lower the minimum detectable radon concentration.	This study found that a four-day exposure yielded the most accurate results. Additionally, the enhanced detection efficiency is enhanced by using two scintillation detectors instead of one [30].
7. Miles, 2001	This study assessed the accuracy of various passive radon detectors and investigated the effectiveness of seasonal correction factors for improving annual estimates.	Researchers assessed detector response (charcoal, etched track, electret) across exposure times (4–90 days), using weighted averages for radon decay and desorption. This analysis considered both the full dataset and one excluding an anomaly.	Longer charcoal exposures (30+ days) yielded more accurate annual averages (a factor of 1.2) than shorter ones (4–7 days), with a difference factor of 2.7 [31].
8. Rey et al., 2023	Evaluated the reliability of a recently introduced short-term radon measurement methodology in Switzerland. This method was intended to assess the risk of exceeding the national reference level (300 Bq/m^3^) under typical occupancy conditions.	To assess the impact of meteorological factors on indoor radon, 120 h active measurements, validated against year-long passives (where available), were conducted at 12 locations using METAS-approved and calibrated sensors with subsequent statistical analysis.	A study found that weather conditions in the preceding five days significantly impacted short-term radon measurements. This effect varied by occupancy, location, and building use [32].
9. Ptiček Siročić et al., 2020.	Compared short-term (4–8 days) indoor radon concentration measurements with previous long-term data (1 year) in northern Croatia.	Radon concentration was continuously measured using Airthings Corentium Pro at 15 randomly chosen locations for one year. Measurements were conducted in residential buildings, office buildings, weekend cottages, and small family wine cellars.	Short-term tests accurately identified homes that were exceeding administrative radon levels for low concentrations (<100 Bq/m^3^), capturing around 80% of long-term (>90 days) radon test data [33].
10. Denman, 2008	Assessed the accuracy of short-term (one-week) radon detectors (track–etch, electret, activated charcoal) compared to long-term (three-month) track–etch detectors for estimating annual radon exposure.	Long-term monitoring: Durridge RAD-7 systems continuously measured radon levels at hourly intervals for a year in three properties. Short-term monitoring: short-term detectors (track–etch, electret, activated charcoal) were placed near the RAD-7 systems for one week.	One-week radon measurement using track–etch/electret accurately predicted annual averages in low/moderate-radon areas (i.e., <75 Bq/m^3^), especially for newly mitigated homes. Three months were recommended for high-radon regions [19].
11. Hull, 1990	Assessed the reliability of short-term (1–7 day) radon measurements for predicting long-term radon exposure.	Continuous radon concentration data were collected for one year in an unmitigated New Jersey basement using a Wrenn chamber.	Short-term measurements predicted the annual average radon level with 50% accuracy. The results were most reliable in winter and spring [34].
12. Li et al., 2023	Evaluated the extent to which a long-term radon measurement can be predicted via a collocated short-term radon measurement under different conditions.	For short-term measurements (2–7 days), the study employed PicoCan-400 test kits, followed by gamma spectrometry to determine the average radon concentration. Long-term measurements were conducted using CR-39.	Short-term radon measurements, combined with other factors, effectively predicted seasonal variations (up to 79%) and annual averages (up to 67%) [35].
13. Al-Jarallah et al., 2008	Investigated the correlation between short-term (24 h) and long-term (6 months) indoor radon measurements at low radon levels.	Short-term measurement: an AlphaGUARD radon gas analyzer was used for 24 h at 34 locations. Long-term measurement: CR-39-based radon dosimeters were deployed for 6 months at the same locations.	Long-term radon levels were 1.3 times higher than short-term levels, indicating that short-term measurements may not reliably show long-term exposure [36].
14. Barros et al., 2014	Investigated the accuracy of short-term (7–10 day) measurements compared to annual radon averages.	Trained personnel conducted short-term measurements (electret ion chamber) for 7–10 days and long-term (alpha track) radon measurements in Iowa basements.	Short-term tests accurately predicted that annual radon exposure would exceed the action level (148 Bq/m^3^) in 88% of cases and exceed 74 Bq/m^3^ in 98% of cases [37].
15. Barros et al., 2016	Assessed the effectiveness of short-term radon measurements in predicting long-term (1 year) residential radon exposure in 158 Iowa residences.	Short-term (7 days) basement measurements were compared with 1-year measurements using alpha track detectors. The indoor radon concentrations were also compared to the US radon action level (148 Bq/m^3^).	Short-term tests identified 44% of residences as exceeding the action level based on year-long living space concentrations [38].
16. Nunes et al., 2023	Compared the quality of the results obtained in the long-term and short-term modes.	Two AirThings Corentium Plus Radon Monitor probes were deployed for a one-month period. Short-term datasets (1 day, 1 week, 1 month) were randomly selected from the merged data.	Short-term monitoring campaigns lasting a month, a week, or a day did not produce a statistically representative description of indoor radon exposure for the monitored building [16].

This study investigated the accuracy of short-term radon tests in predicting long-term radon exposure and the results are shown in Table 2. Our review findings revealed that the effectiveness of these tests in predicting long-term radon concentrations depended on several factors, including, measurement duration, season, baseline radon concentrations, meteorological conditions, and inhabitants’ behavior. One-week indoor radon measurements were accurate in predicting seasonal and annual indoor radon concentration in low-risk areas (where the indoor radon concentration is below 75 Bq/m^3^) and new buildings. Furthermore, one-week radon measurements effectively predicted seasonal indoor radon concentrations with 79% accuracy. However, a longer measurement duration (a month or more) was generally preferred, especially in high-risk zones, in obtaining a more precise representation of annual averages. For indoor radon-screening purposes and to monitor adherence indoor radon exposure reference/action levels, a minimum of 4 days appeared to be necessary to obtain reliable results that were consistent with seasonal readings. Furthermore, winter month radon measurement yielded more accurate results that were consistent with the annual results. The findings of this review revealed that considering both prevailing weather conditions in the days preceding the test and the dwelling’s occupancy patterns improved the accuracy of short-term radon measurements. 

## 4. Discussion

### 4.1. Radon Measurement and Regulation in Europe and US

#### 4.1.1. Europe

Geological, environmental, and occupant factors cause significant fluctuations in radon levels, necessitating long-term measurement (exceeding 3 months), which is the preferred method or gold standard for accurate prolonged exposure assessment [5,37,39]. Radon levels can fluctuate by a factor of 2 to 3 within a 24 h period, with the highest levels typically occurring during the night and early morning [12,40,41,42]. Regulatory bodies such as the ICRP, IAEA, and WHO support this approach, with most international standards requiring monitoring periods exceeding 3 months [5,6,42,43]. The International Organization for Standardization (ISO) recommends a minimum of 2 months, while the European Basic Safety Standards Directive 2013/59/EURATOM (hereafter EU-BSS) mandates year-long measurements in order to monitor compliance with the 300 Bq/m^3^ reference level for both new and existing dwellings [44,45]. Although twelve-month radon measurements are considered the most accurate for radon exposure assessment due to their ability to capture seasonal and other fluctuations [5,12,43], often, seasonal (3 months) and semester (6 months) radon measurements, together with the use of seasonal correction factors (SCFs), are used to predict the long-term radon average [27]. Several European countries have adopted stricter national reference levels, ranging from 100 Bq/m^3^ (Denmark, the Netherlands, and Norway) to 200 Bq/m^3^ (United Kingdom, Italy, Ireland, Estonia, Finland, and Sweden) [44,46,47,48,49,50]. Adherence to the prescribed reference levels for radon is often non-mandatory in most EU member states [51]. The long-term monitoring of radon levels is vital for the development of radon maps and the identification of Radon Priority Areas (RPAs) as required by the EU-BSS [49]. Radon Priority Areas (RPAs) are zones with a potentially significant number of dwellings exceeding safe radon levels. However, the definition of “significant” varies significantly across Europe [45,48,49,50,51,52,53]. Some European states (Ireland, Spain and the UK) require that, after identifying RPAs, all new homes built in RPAs be built with measures and materials to protect against radon [44,54]. This inconsistency is problematic due to the inherent spatial variability of radon [55,56]. Elevated levels can even occur outside RPAs, creating a potential blind spot in the European strategy that prioritizes RPAs for radon management [49,57,58,59]. Regardless of RPA designation, long-term measurements in individual homes are essential for accurate risk assessment, which involves evaluating hazard, susceptibility, and exposure [49,58,59]. A detailed comparison of European and US radon management strategies is presented in Table 3. This table allows for a clear visual analysis of the approaches taken by each region. 

#### 4.1.2. United States (US)

The Environmental Protection Agency (EPA), established in line with Superfund Amendments and Reauthorization Act (SARA) of 1986, has been tasked with assessing radon health risks and mitigation strategies in the US [24]. The Indoor Radon Abatement Act (IRAA) of 1988 further empowered the EPA to conduct radon studies and identify high-risk areas like dwellings [23]. Based on numerous national surveys conducted, the EPA has set an action level (AL) of 148 Bq/m^3^ for indoor radon, prompting mitigation efforts in homes exceeding this level [35]. While the EPA’s radon map classifies counties by predicted radon risk (Zone 1: high; Zone 2: medium; Zone 3: low), the EPA recommends testing all homes regardless of the predetermined classification [35,56,60]. Public awareness is promoted through the Federal Radon Action Plan (FRAP) launched in 2011. FAP’s goals include unifying federal radon programs, educating the public, and incentivizing mitigation. Currently, 34 states have enacted radon-related laws, with 29 requiring radon disclosure during real estate transactions and nine mandating radon-resistant features in the construction of new dwellings [60]. The EPA recommends radon testing in all homes before purchase or sale and offers resources like “A Citizen’s Guide to Radon” [13,61]. Additionally, twelve US states mandate certification for radon service providers and require them to submit test results to state agencies [62]. The US aims to reduce indoor radon to outdoor levels, aligning with the American Lung Association’s National Radon Action Program (NRAP), which aims to mitigate radon levels in millions of homes and prevent thousands of lung cancer deaths annually [22]. The US EPA advocates for a graded approach via rapid STT, which is followed by confirmatory long-term tests only in exceptional cases [1,63,64]. This strategy balances efficiency and accuracy, with the averaging of multiple STT results across natural fluctuations enhancing accuracy before costly mitigation decisions [63,64]. This approach has empowered homeowners with greater confidence in addressing radon exposure [63,64,65]. In the US, STT (2–5 days) dominates, particularly during real estate transactions, despite the limitations in capturing long-term exposure [1,35,37]. The EPA promotes affordable STT kits to equip the public with knowledge about radon and enable widespread participation in identifying high-risk areas and directing mitigation efforts [1]. This citizen science approach necessitates that researchers ensure data accuracy through independent performance evaluations [1]. The US utilizes a tiered system for radon assessment and mitigation in homes. The initial step involves an STT (2–10 days) conducted under closed-building conditions in the lowest occupied level [24]. A follow-up test is recommended if the initial result exceeds the action level of 148 Bq/m^3^. Similar to Europe, the EPA recommends using two devices for simultaneous testing [63,64]. The type of follow-up test depends on the initial results. High initial results necessitate a second STT, while moderate results allow for either a short-term or long-term (over 90 days) follow-up test. Long-term tests provide a more accurate picture of annual radon levels, but STT offers faster turnaround times for mitigation decisions. Mitigation is recommended when long-term follow-up results exceed 148 Bq/m^3^ or the average of the initial run and when the second STT exceeds the action level [24]. STT serves as a screening tool, while long-term tests provide a more diagnostic function. Common short-term detectors include charcoal canisters, alpha track detectors, and continuous monitors [13,60]. Technological advancements, such as affordable continuous monitors, hold promise for further expediting the radon testing process and potentially encouraging mitigation efforts [66]. The rise of citizen science in radon testing necessitates that researchers ensure data accuracy through independent radon tests conducted by homeowners [1]. While consumer grade detectors and real-time smart sensors offer exciting possibilities for instant monitoring, their accuracy requires rigorous investigation before widespread adoption [19,25,67]. A detailed comparison of European and US radon management strategies is presented in Table 3. This table allows for a clear visual analysis of the approaches taken by each region. 

**Table 3 sensors-24-04575-t003:** Comparison of radon management strategies in Europe and US.

Aspect	Europe (Long-Term Radon Testing)	United States (US) (Short-Term Radon Testing)	References
**Authorities Responsible**	The European Union provides the overall framework for radon control in Europe, and individual member states are responsible for implementing specific radon control measures within their own territories.	The oversight of radon activities in the United States is the responsibility of the Environmental Protection Agency (EPA) at the federal level, with individual states also contributing to regulating and managing radon-related issues.	[5,13,49]
**Radon Policies**	While national reference levels for radon vary across Europe (100 Bq/m^3^–300 Bq/m^3^), the EU Basic Safety Standards Directive (2013/59/EURATOM) and the Atomic Law Act, amended in 2019, sets a maximum level of 300 Bq/m^3^, with some countries adopting stricter national guidelines.	The Indoor Radon Abatement Act (IRAA) of 1988 established an action level of 148 Bq/m^3^ (4 pCi/L) to trigger radon mitigation efforts.	[5,22,35,46]
**Strategies (Existing Buildings)**	Radon testing is recommended, with a focus on Radon Priority Areas (RPAs). Some countries now require radon-resistant features in new construction, and mitigation is recommended when reference levels are exceeded.	Short-term radon testing is recommended before buying/selling a home, and mitigation is recommended when the action level is exceeded.	[13,44,49]
**Strategies (New Buildings)**	Fewer states (Ireland, Spain, and the UK) require radon-resistant features in new homes.	More states in the US (more than 9 member states) mandate radon-resistant features in new constructions.	[44,60]
**Measurement Approach**	Long-term measurements (at least 3 months, preferably 12 months)	Short-term tests (2–10 days), followed by a confirmatory long-term test (optional) if initial results exceed action level.	[5,13,24]
**Devices Used**	Passive: alpha track detectors and electret ion chambers. Active: continuous radon monitors	Passive: charcoal canisters and alpha track detectors. Active: continuous radon monitors.	[5,12,13,68]
**Radon Measurement Strategy Benefits**	Provides a more accurate year-round picture of radon levels for risk assessment, capturing seasonal variations in radon concentration, aligned with the recommendations of international organizations such as WHO and IAEA for risk assessment. Often, two radon detectors are used per dwelling.	The US short-term radon strategy is associated with faster turnaround time for results (days vs. months), greater cost-effectiveness for initial screening, and easier implementation for large-scale testing (e.g., real estate transactions).	[5,6,35,63]
**Radon Measurement Strategy Limitations**	Requires longer deployment times, delaying mitigation decisions. May not be practical in all situations (e.g., real estate transaction and tenant occupancy). The reliance on seasonal correction factors (SCFs) may introduce uncertainties.	Less accurate for capturing long-term exposure patterns. May underestimate radon levels during periods of high ventilation. May overestimate radon levels during colder months with closed-house conditions. Similar to Europe’s approach, two detectors are used per dwelling.	[5,27,64]
**Radon Risk Zones**	The EU emphasizes Radon Priority Areas (RPAs) based on geological predictions. Individual home testing remains crucial, regardless of RPA designation.	Radon risk zones are displayed on the National Radon Map, showing predicted risk levels (high, medium, and low). Testing is recommended for all homes, regardless of their radon risk zone.	[22,48,49,56]
**Building Codes**	Building codes vary in their implementation across member states. For example, some countries such as Ireland, Spain, and the UK require radon-resistant features in new constructions.	In the United States, around 9 states (i.e., Delaware, Florida, Iowa, etc.) mandate radon-resistant features in new constructions.	[44,54]
**Radon Awareness**	The primary focus is on encouraging voluntary testing. Discussions about mandatory testing are ongoing.	Public awareness campaigns through federal initiatives (e.g., Federal Radon Action Plan (FRAP)) are conducted annually.	[44,60]
**Dwelling Coverage**	Individual homeowners or public initiatives drive radon testing, but radon data is incomplete due to reliance on voluntary testing. Less than 1% of the buildings in Europe has been tested for radon	Large-scale testing is often conducted during real estate transactions. The US has witnessed a paradigm shift in radon testing practices, with over 23 million short-term tests (98% of the total) conducted.	[1,24,69]
**Collective Risk Assessment**	While long-term radon testing remains common for individual risk assessment, its value in population-level risk estimation is limited by sample size, representativeness (i.e., volunteer bias, high drop-out rate), and potential underestimation due to incomplete voluntary testing and variable measurement periods.	Provides a more comprehensive picture of national radon exposure patterns. Does not capture long-term risks due to reliance on STT. Conflicting evidence regarding the correlation between short-term and long-term levels complicates their use in evaluating health risks.	[1,35,36,37,38,39,40,41,42,43,44,45,46,47,48,49,52,70]
**Average radon concentrations**	98 Bq/m^3^	46 Bq/m^3^ in US	[5,22]

#### 4.1.3. Challenges Facing Radon Management Strategies

Radon management strategies face significant challenges, particularly in terms of the reliance on voluntary public testing and mitigation [44]. Public awareness campaigns about radon have failed to improve public knowledge, resulting in a knowledge–action gap due to the underestimation of health risks [49,71,72,73]. Force et al. emphasize the need for public awareness campaigns to combine risk education with easy access to testing kits to encourage action [74]. In Europe, online platforms for radon information are currently inadequate and lack engagement [44]. Current legislation in Europe has not significantly increased the radon testing rate [45]. Even though building codes in some EU and US member states incorporate radon protective measures, these measures do not eliminate the need for radon testing [74]. Furthermore, mandatory post-construction testing is largely absent across the EU [44,45]. This, combined with the potential for radon levels to rise above reference levels after mitigation, highlights the need for a more comprehensive approach [75,76]. 

Canada’s and US successful strategy, which utilizes free guidelines for self-radon testing and self-remediation, may provide a valuable model for Europe [68]. This approach could incentivize non-mandatory testing and remediation. While targeting high-risk areas with public education and outreach efforts would optimize resource allocation, it is important to note that elevated indoor radon concentration can be found anywhere, regardless of the risk zone [22,77]. Financial constraints pose another hurdle to the progress of radon initiatives. Limited funding discourages testing across the EU, and a shortage of qualified radon mitigation professionals restricts the options available to the public [45,78]. The literature suggests that the budget of current EU member states is inadequate for conducting nationwide radon surveys using long-term radon measurement, hence the need to adopt a cheaper approach. The use of short-term radon measurement seems to be an appropriate alternative [1,2,78]. The successful registration and training programs for radon professionals in Ireland and Canada serve as examples [25,62]. Furthermore, integrating radon testing into real estate transactions, as seen in parts of the UK and US, could lead to more radon tests [25,78,79]. Moreover, there is a persistent lack of awareness not only among the public but also crucially among decision-makers, healthcare professionals, and architects, even in high-risk areas; this issue requires attention [44,45]. Inconsistencies in state-level radon reference levels and radon testing data hinder national estimations in the US and Europe [44,45,74]. Data coverage varies due to resource allocation disparities, and incomplete databases plague many states. Internal data resolution is often hampered by the lack of address-level information and the inability to distinguish pre-mitigation from post-mitigation tests, leading to inflated testing rates and skewed analysis in the US [74]. 

#### 4.1.4. The Feasibility of the US Approach in the EU 

Indoor radon testing regulations and protocols differ significantly between Europe and the US. The US adopted STT and action levels (AL) of 148 Bq/m^3^, while Europe utilizes reference levels (RL) ranging from 100 to 300 Bq/m^3^ and mandates long-term testing (minimum two months). Notably, the AL is not a direct equivalent of the RL [1,5,57]. Statistics suggest that the US approach has been more efficient in terms of dwelling coverage [57]. However, simply adopting the lower US AL in Europe would necessitate remediating many dwellings compliant with the current 300 Bq/m^3^ threshold [80]. This could be financially burdensome, considering Europe’s already higher average radon concentration (98 Bq/m^3^) compared to the US (46 Bq/m^3^) [5,22]. European countries could adopt a balanced approach, integrating short-term screening tests for initial assessments (e.g., property transactions) in low-risk areas, followed by confirmatory long-term testing for high-risk or inconclusive cases [57]. This may optimize cost-effectiveness and accuracy. Europe could significantly increase testing rates by adopting mandatory testing during property transactions and continuous public awareness campaigns like Canada’s “100 Test Kit Challenge” and the US’s Radon Action Months [21,23]. Furthermore, public educational efforts should leverage the internet and best practices like customized information websites and social media engagement [44,45,81]. EU member states might benefit from a shift towards decentralized radon campaigns, empowering public participation in awareness efforts, as seen in successful programs in Iowa and Canada [20,21]. Concerns regarding accuracy and affordability can be addressed by recognizing that the average global radon level (30–39 Bq/m^3^) often falls below reference limits, meaning that repeat testing in borderline cases, using either short-term or long-term methods, may not be necessary in most instances [18,20,57,58,82,83]. A comprehensive approach that incorporates the strengths of both European and US strategies is essential. By adopting a balanced testing approach, leveraging public awareness campaigns, and fostering public participation, European countries can significantly increase radon testing rates and improve public health outcomes. 

### 4.2. The Usefulness of Different Radon Testing Strategies in Collective Risk Assessment

Aligned with international agencies like the IAEA, European radon policies prioritize identifying Radon-Prone Areas (RPAs). This approach often misses high-radon pockets outside designated RPAs due to spatial variations in radon levels [18]. This challenges the assumption of safety in non-RPA zones and emphasizes the need for additional strategies in order to comprehensively assess and reduce the health risks associated with indoor radon exposure [49,56,57,84]. While long-term radon measurements have traditionally been relied upon in studies of radon-induced health problems, recent research questions the effectiveness of relying solely on long-term testing for collective or population risk assessments [1,49,57]. The argument is that accurate risk estimation depends on a well-designed sampling strategy, encompassing a representative range of buildings (representative sample), rather than solely on the extended duration of individual measurements [1,2,27]. While long-term radon testing provides a more comprehensive picture of exposure by minimizing seasonal variations [5,12,43], its effectiveness in population-level risk assessment is hampered by limitations in sample size, representativeness, and participation rates [31,35,48,52]. Additionally, the logistical challenges associated with long-term deployments, including cost, time demands, and inconvenience for residents and professionals, further hinder its widespread application [24,29,35,49,65,70]. Europe’s meticulous long-term testing, while thorough, yielded limited impact. Testing rates remained low (less than 1% of buildings), and estimates suggest that only 26,000 homes were remediated out of the millions likely needing attention [1,6,75]. The low coverage in terms of testing dwellings has spurred discussions among radon professionals regarding mandatory testing and the use of STT in specific situations, like real estate transactions and mitigation verification [44,45]. This mirrored observations in the US, where long-term testing represents only 2% of all radon tests conducted [69]. The limited number of long-term radon tests was attributed to a lack of knowledge, resulting in few individuals being willing to conduct such tests [1,45,57,61]. The fast-paced nature of modern times makes it challenging for end-users and service providers to perform long-term radon testing due to time constraints [24,85]. Recognizing the limitations of traditional long-term radon testing methods, La Verde et al. (2019) proposed a multi-pronged approach that included public education on radon risks, emphasizing survey importance, and confirming detector receipt [86]. Europe’s long-term radon testing has been questioned for its efficacy in monitoring compliance with radon reference levels and addressing public health concerns in the context of millions of untested and unmitigated buildings [1,2,35,37]. Despite minimal disruption and prolonged exposure assessment, alternative strategies are needed to efficiently evaluate radon exposure in Europe’s vast building inventory. This is further supported by the fact that the accuracy of risk estimation depends on well-designed sampling strategies, rather than the length of individual measurements [1,2,57]. Traditional long-term radon surveys often suffer from representativeness issues due to non-random sampling methods such as volunteer participation, which can introduce bias as volunteers are more likely to suspect radon issues in their homes [38,48,52]. This underscores the need for a new approach that prioritizes a representative sampling strategy, assessing population risk by covering many buildings.

Although STT accuracy for long-term exposure is debated, it remains valuable for initial screening and community assessments, particularly in low-risk areas. The use of STT in health studies or exposure assessment is limited by uncertainties in predicting long-term exposure [1,35,36,37,85]. Efficient screening procedures are crucial to identifying buildings that exceed the reference level mandated by paragraph 3.46 of IAEA SSG-32 [87], as millions globally require radon assessment. At present, STT appears to be a more dependable option for the extensive screening and detection of buildings exceeding prescribed radon levels than the conventional radon measurement techniques utilized in Europe [55]. The United States serves as a successful example, with over 23 million short-term tests (98% of the total) conducted compared to only 500,000 long-term tests (2%) [69]. With short-term radon testing, the US conducts 30–50 times more tests compared to Europe [1]. The widespread adoption of short-term radon testing, driven by its affordability and ease of use, has led to the mitigation of over 1.24 million homes based on STT results [2,69]. Short-term radon tests have enabled US homeowners to proactively address radon exposure risks. Funding for testing and mitigation has come from various sources, including the EPA, mortgage insurance, occupants, and property owners [11,22]. STT are particularly valuable when considering exposure at the population level. The large datasets generated by these tests allow for robust statistical analysis, reducing individual uncertainties and providing reliable estimates of radon exposure for entire communities [35]. This approach may pave the way for comprehensive radon maps, targeted mitigation efforts, and ultimately, improved public health. While short-term radon measurements excel at initial screening and community-level assessments, their application in health studies remains hampered by uncertainty about their accuracy in predicting long-term exposure. 

### 4.3. Short-Term Measurement Accuracy in Predicting Long-Term Exposure

Radon exposure is a significant public health concern worldwide, and the accurate measurement of indoor radon concentrations is crucial for effective mitigation strategies. Long-term testing methods, typically lasting for three months or more, have been the gold standard for assessing annual average radon exposure and individual risk. Short-term radon measurements are not typically considered by the IAEA, ICRP, WHO, or by member states of the European Union [22]. However, traditional long-term radon testing methods are associated with numerous challenges, leading to low participation rates. Consequently, there is growing interest in the feasibility of short-term radon testing (STT) as an alternative to rapid screening and population-level risk estimation [22]. This section of the review critically analyzed the strengths and limitations of STT for predicting long-term radon exposure, drawing upon relevant research findings. This section addressed the critical question of how closely STT corresponds to the more comprehensive picture provided by long-term monitoring [35,36,37]. Numerous studies have investigated the precision of short-term testing (STT) in comparison to long-term measurements. Several studies (8 studies) indicated that STT can serve as a dependable indicator of annual averages under specific conditions [19,26,28,29,33,35,37,38]. For example, research by Miles et al. (2004) and Denman (2008) illustrated that track–etch or electret radon readings over a one-week period falling below 75 Bq/m^3^ could reliably predict that the annual average is below the action level in regions with low-to-moderate radon levels [19,28]. Additionally, Barros et al. (2014) discovered that STT accurately predicted that long-term radon averages would exceed the action level in 88% of instances [37]. However, several studies have noted constraints. Notably, Hull (1990) proposed an approximate maximum accuracy of 50% for predicting annual averages using short-term methods, emphasizing the inherent variability in radon concentrations [34]. The results suggest that short-term tests are most appropriate in regions with low radon levels, such as those near the global indoor radon average of 39 Bq/m^3^ [1,5,22]. STT results indicating low indoor radon levels can be deemed reliable, especially when values are close to the global average of 30–40 Bq/m^3^ [1,2]. Even with a 100–200% margin of error, STTs in such cases would not surpass the regulatory limits, making them a cost-effective and efficient initial assessment [2,57]. Miles et al. 2004 observed that if STT reveals significantly higher values, like 500 Bq/m^3^, such results should be treated as conclusive evidence that the reference limit has been exceeded [28]. A study conducted in the United States by Li et al. found that although STT effectively predicted the seasonal radon average, STT showed moderate ability in predicting annual radon levels [35]. Li et al. further proposed that averaging measurements from all four seasons within a building provided a more accurate estimate of annual concentrations for chronic health studies [35]. Al-Jarallah et al. (2008) noted that short-term radon testing tends to underestimate long-term averages, especially at low radon levels [36]. This finding was consistent with the argument made by Bochicchio et al. (1995) regarding the challenges associated with accurately measuring radon variations [3]. These variations occur over different time intervals, ranging from hourly to annually, and are influenced by seasonal factors, weather, building characteristics, and occupant behavior. 

This discrepancy could be due to variations in the radon detectors used, measurement durations, geographical locations, and environmental conditions across different studies. Furthermore, the studies consulted suggested that the accuracy of short-term radon test is influenced by several factors, namely duration, season, building characteristics, and the radon detector used. The optimal duration for an STT remains under debate. While some research suggests the need for a minimum of four days of assessment for robust data analysis and the accurate prediction of long-term radon average [29,35], others propose one week for initial screenings in low-risk areas [26]. Notably, high-risk zones may require extended testing periods, potentially lasting up to three months [26]. Studies have consistently reported improved accuracy with longer STT durations (e.g., one week or more) compared to very short deployments (less than four days) [26,30,33]. Radon concentrations fluctuate throughout the year, with winter months often exhibiting higher levels [26,27]. Seasonal variations in radon concentration can significantly impact the accuracy of short-term testing (STT) [34]. Winter and spring months typically exhibit the levels closest to the annual average, suggesting that strategically timed STTs during these periods might yield more reliable results [26]. In other words, STTs conducted in winter may provide a more accurate picture of annual exposure compared to summer measurements. A study by Rey et al., 2023, found that building characteristics such as airtightness, occupancy patterns, and location within the building (basement vs. upper floors) can significantly influence radon levels [32]. The choice of detector technology also plays a crucial role. While various STT detector options exist (e.g., electret ion chambers, CR-39 detectors, and activated charcoal canisters), their accuracy varies [27,31]. Studies have shown that etched track detectors and electrets with longer exposure periods (90 days) tend to outperform other methods in terms of accuracy [31]. Ideally, STT should account for these factors to achieve better interpretation. 

While STTs excel at reflecting seasonal radon fluctuations [35], their accuracy weakens when capturing year-long exposure. Knowing the temporal variation in indoor radon concentration helps to identify the best time to test indoor radon concentration to avert underestimating the risk of prolonged radon exposure [88]. To address issues arising from the growing concern of temporal variability associated with short-term radon measurement, the US targets months (the fall and winter months) in which indoor radon is expected to be high and when people are likely to spend most of their time indoors [88]. The literature suggests that indoor radon measurements taken during winter do not show significant temporal variability; hence, the metric of temporal correction factors (TCFs) is often not applied [85]. In addition, to compensate for seasonality when assessing radon risks, short-term measurement outcomes are generally converted to equivalent mean annual levels by the application of a seasonal correction factor (SCF) [89]. Concerns over the use of the SCFs of a geographical or geologically diverse region are currently a contentious issue [85,89]. Temporal variation, a natural fluctuation in radon levels, affects both short-term and long-term testing methods. While STTs may be more susceptible due to their limited sampling window, year-long measurements are not exempt [3]. Even studies spanning a decade observe significant year-to-year variations, with one study reporting a 17% fluctuation [31].

## 5. Future Direction and Conclusions

Successful radon testing programs hinge on the use of a multi-pronged approach. This includes fostering community engagement through incentives, building partnerships across sectors, and utilizing targeted public education campaigns. By implementing these measures alongside standardized testing methodologies, public health officials can effectively raise awareness and encourage radon testing, ultimately safeguarding public health. The battle against radon exposure demands the accurate assessment of risk, a task complicated by the inherent trade-offs between long-term and short-term testing methods [2]. This analysis dissected the strengths and limitations of each approach, paving the way for a strategic blend in order to optimize radon management. Long-term measurements, championed by Europe, boast unparalleled accuracy in capturing annual exposure patterns. Their year-long duration minimizes temporal uncertainties, ideal for individual risk assessment and mitigation decisions [6,42]. However, their resource-intensive nature often translates to low coverage rates and limited mitigation outcomes, raising questions about their overall efficacy. The reviewed studies suggested that STT can be a reliable predictor of annual radon exposure and may be useful in determining compliance with radon reference or action level. Therefore, a paradigm shift is necessary, moving beyond a rigid adherence to either method. A two-stage approach may provide a reliable solution, initially requiring no less than 4 days of screening, in order to identify high-risk areas quickly and cost-effectively. This is followed by confirmatory long-term tests in borderline cases for precise data acquisition before making mitigation decisions. This strategy leverages the strengths of both methods, maximizing accuracy and resource allocation.

Moving forward, the following principles should guide radon management:

It is necessary to prioritize representative sampling over extended measurement durations. Long-term tests alone, with their low coverage rates, may not be a panacea. Integrating radon data collected from quick, short-term tests with strategically aggregated data will produce a more thorough assessment of population-level risk. There is a growing body of knowledge advocating a graded approach to radon testing in Europe, which will entail the use of short-term radon testing for screening purposes, and if reference levels are exceeded, then long-term radon testing may be warranted [45,47].

Promoting proactive testing initiatives, especially during real estate transactions, can significantly increase testing rates and empower individuals to manage their radon exposure risks in Europe. 

Research on short-term test accuracy is essential for improving reliability in health studies and optimizing mitigation strategies by understanding the correlation between short-term and long-term exposure.

## Figures and Tables

**Figure 1 sensors-24-04575-f001:**
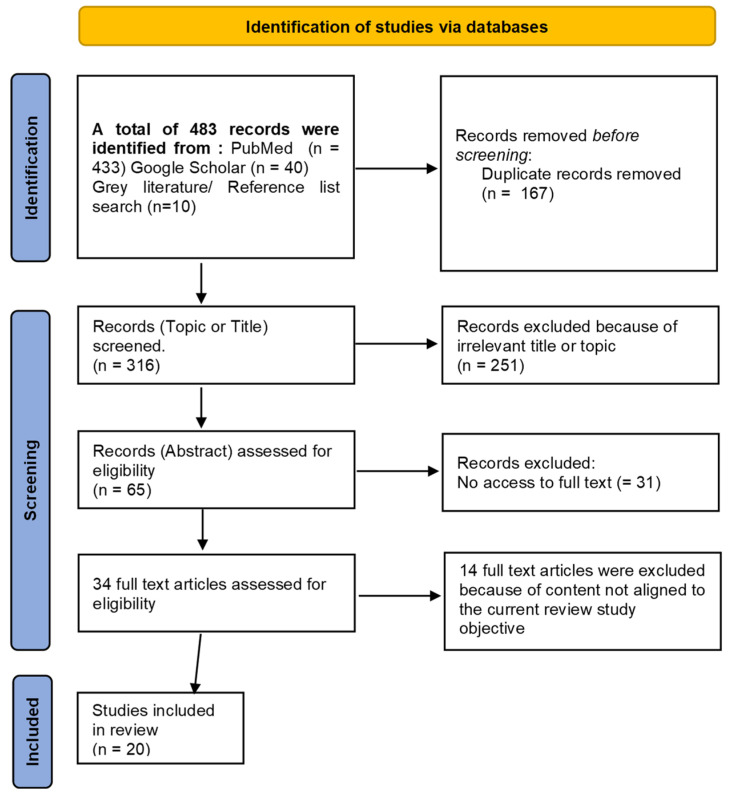
Flow diagram of literature search.

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
