# Peer review of "Short-Term vs. Long-Term: A Critical Review of Indoor Radon Measurement Techniques"

_sensors, 2024, doi:10.3390/s24144575_

Round 1
Reviewer 1 Report
Comments and Suggestions for Authors
Ms is well written and interesting. There is comparison between US and European practice in radon measurements. Also comparison between short term active and long term passive radon measurements is presented. Literature survey is comprehensive and well presented.
My only remarks is that ms too long and too wordy. Let author attempt to shorten it in some points without loosing the content.
Author Response
Subject: Response to reviewer’s comments
Reviewer 1 Comments
Ms is well written and interesting. There is comparison between US and European practice in radon measurements. Also comparison between short term active and long term passive radon measurements is presented. Literature survey is comprehensive and well presented.
My only remarks is that ms too long and too wordy. Let author attempt to shorten it in some points without loosing the content.
Author’s response:
Thank you for your positive feedback on our manuscript. We appreciate your acknowledgment of its well-written nature, the interesting comparison of US and European radon measurement practices, and the comprehensive literature review.
We agree with your suggestion regarding the manuscript's length. We have carefully reviewed the manuscript and successfully condensed it from 22 pages to 19 pages in the revised version. This conciseness was achieved by summarizing various sections while ensuring the core content remained intact.
Specifically, we have streamlined the following sections: Line 41 -43, 51-53, 61 – 66, 78, 152-155, 183 -187, 244- 247, 250 – 257, 259 – 260, 265 – 267, 285 – 287.
Additionally, we have condensed the information presented in Tables 1 and 2 (Lines 104 and 111, respectively) to improve conciseness. Furthermore, we have removed extraneous details from Lines 87-89 and Line 149.
We believe these revisions enhance the manuscript's readability and focus without sacrificing crucial information. We appreciate your valuable feedback and hope the revised version meets your expectations.
Sincerely,
The Authors
Reviewer 2 Report
Comments and Suggestions for Authors
This review evaluated the accuracy of STT in predicting annual radon averages and compared testing strategies in Europe (where long-term measurements are common) and the United States (where STT is prevalent). Twenty (20) studies were systematically identified through searches of scientific databases and gray literature, focusing on STT accuracy and radon management. The manuscript uses fluent English and sotry yelling which helps the reader fully understand the text.
The aim is clear and easily intelligible, however it may not meet with the unanimous favor of sector experts, as carrying out very short measurements requires a profound evaluation of the environmental and structural parameters of the places where the measurements are carried out.
The reflection input is instead very useful to the scientific community to increase the commitment to finding standardized models of radiometric characterization of sites regardless of their geographical and administrative location, given that the risk is the same for humans.
Author Response
Reviewer 2 Comment
The aim is clear and easily intelligible. However, it may not meet with the unanimous favor of sector experts, as carrying out very short measurements requires a profound evaluation of the environmental and structural parameters of the places where the measurements are carried out.
Response:
We thank the reviewer for the valuable feedback and appreciation of the clarity of our critical review's aim. We understand the concern regarding the acceptance of short-term radon measurement techniques within the sector, particularly the necessity for a thorough evaluation of environmental and structural parameters.
We acknowledge the importance of considering environmental and structural parameters when conducting short-term radon measurements. Our review already includes a discussion on how these factors can influence measurement accuracy and reliability. Also, our review provides a comprehensive evaluation of both short-term and long-term radon measurement techniques to provide a balanced argument when considering both. For short-term methods, we emphasize the need for careful site assessment and the conditions under which these methods can yield reliable results. Short-term radon measurements offer significant advantages, such as quick initial assessments and the ability to identify potential radon hotspots, and these advantages become crucial in epidemiological studies and when performing health risk assessments following potential radon exposures.
Thank you.
Reviewer 3 Report
Comments and Suggestions for Authors
This article is an excellent research report, but the main content of the paper is not related to the radon detection and measurement techniques, thus I don't think the research in the article matches the SCOPE of this journal. I have to suggest reject for this paper, it should be submitted to other journals.
Comments on the Quality of English LanguageThe English is good to me.
Author Response
Reviewer 3 comment
This article is an excellent research report, but the main content of the paper is not related to radon detection and measurement techniques.
Response:
We appreciate the reviewer's positive feedback on the quality of our research report. However, we respectfully disagree with the assertion that the main content of the paper is not related to radon detection and measurement techniques, thus, not within the journal scope. Our manuscript extensively discusses and addresses various aspects of radon detection and measurement as follows:
We provide a comprehensive critical review of radon measurement techniques, particularly focusing on the short v/s long-term measurement strategies, establishing the necessity of effective detection and measurement techniques. This falls within the scope (Novel detectors and monitors for radon) of the journal’s special issue on “Detection and Measurement of Radioactive Noble Gases”. Within our article, we strongly highlight the advantages and limitations of the short- and long-term detection techniques, which underscores the relevance of both techniques within the current scientific context in terms of measurement sensitivity, accuracy, and applicability.
We believe that the core of our research article is indeed centered around radon detection and measurement techniques. Which in our case, we explored techniques and detection methods for both short- and long- terms.
Thank you.
Round 2
Reviewer 3 Report
Comments and Suggestions for Authors
1. This paper has done a lot of research and lists the results of many others, but your own results are not clearly summarized, and it is recommended that the main conclusions of the paper be summarized in Sec3.1 and Sec3.2, respectively.
2. Capitalize the initial letters of abbreviations, and please make them uniform in your article. For example in line 31: Indoor Air Quality (IAQ).
3. Please standardize the expression of units, in some places use Bq/m3 while in some places use Bq*m-3.
Comments on the Quality of English LanguageThe English looks good to me.
Author Response
Mphaga Khathutshelo Vincent
Environmental Health PhD Student
University of Johannesburg
Gauteng
South Africa
+2776 7677 856
+2781 7305 653
Date: 11 July 2024
MDPI Sensors
Special Issue: Detection and Measurement of Radioactive Noble Gases
Subject: Response to reviewer’s comments
Round 2
Comment
- This paper has done a lot of research and lists the results of many others, but your own results are not clearly summarized, and it is recommended that the main conclusions of the paper be summarized in Sec3.1 and Sec3.2, respectively.
Response
Thank you for the valuable feedback. We have incorporated your suggestion by adding dedicated results summaries within sections 3.1 and 3.2. In Section 3.1 (following Table 1), a new paragraph has been included that summarizes the key factors for successful radon testing program implementation, as identified in the analysis (lines 105-111). Similarly, in Section 3.2 (following Table 2), a new paragraph has been included that summarizes the findings on the effectiveness of short-term radon testing for predicting long-term exposure (lines 120-132). We have also added line 367 – 370 to incorporate strategies which could increase radon testing rate in our conclusion
Comment
- Capitalize the initial letters of abbreviations, and please make them uniform in your article. For example in line 31: Indoor Air Quality (IAQ).
Response
We've addressed the inconsistency in abbreviation capitalization throughout the manuscript. All acronyms are now preceded by capitalized terms, ensuring consistent formatting. For example, "Short-Term Testing (STT)" can now be found on lines 13, 59, 64, 118, 120, 178, 187, 189, 190, 23, 267, 283,287, 292, 311, 317, 323, 337, 348, 376. See line 44 for Reference Level (RL) and Action Level (AL)
Comment
- Please standardize the expression of units, in some places use Bq/m3 while in some places use Bq*m-3.
Response
To ensure clarity and uniformity, we have standardized the unit expression throughout Tables 2 and 3. This correction can be found on lines 120 (Table 2) and 200 (Table 3). We appreciate your attention to detail, which has helped improve the manuscript's accuracy and readability.
Sincerely,
The Authors